# Coding of Glioblastoma Progression and Therapy Resistance through Long Noncoding RNAs

**DOI:** 10.3390/cancers12071842

**Published:** 2020-07-08

**Authors:** Alja Zottel, Neja Šamec, Alja Videtič Paska, Ivana Jovčevska

**Affiliations:** Medical Centre for Molecular Biology, Institute of Biochemistry, Faculty of Medicine, University of Ljubljana, Vrazov trg 2, 1000 Ljubljana, Slovenia; alja.zottel@mf.uni-lj.si (A.Z.); neja.samec@mf.uni-lj.si (N.Š.)

**Keywords:** glioblastoma, noncoding genome, biomarker, brain cancer, prognosis

## Abstract

Glioblastoma is the most aggressive and lethal primary brain malignancy, with an average patient survival from diagnosis of 14 months. Glioblastoma also usually progresses as a more invasive phenotype after initial treatment. A major step forward in our understanding of the nature of glioblastoma was achieved with large-scale expression analysis. However, due to genomic complexity and heterogeneity, transcriptomics alone is not enough to define the glioblastoma “fingerprint”, so epigenetic mechanisms are being examined, including the noncoding genome. On the basis of their tissue specificity, long noncoding RNAs (lncRNAs) are being explored as new diagnostic and therapeutic targets. In addition, growing evidence indicates that lncRNAs have various roles in resistance to glioblastoma therapies (e.g., MALAT1, H19) and in glioblastoma progression (e.g., CRNDE, HOTAIRM1, ASLNC22381, ASLNC20819). Investigations have also focused on the prognostic value of lncRNAs, as well as the definition of the molecular signatures of glioma, to provide more precise tumor classification. This review discusses the potential that lncRNAs hold for the development of novel diagnostic and, hopefully, therapeutic targets that can contribute to prolonged survival and improved quality of life for patients with glioblastoma.

## 1. The Noncoding Genome

Only about 2% of the transcribed human genome is composed of protein-coding transcripts, which defines the vast majority of transcripts as non(-protein)-coding RNAs (ncRNAs) [1,2]. Although it was previously thought that ncRNAs are “transcriptional noise”, it has now become apparent that they are involved in different biological and pathological processes [3]. ncRNAs are broadly classified as housekeeping ncRNAs or as those that are ubiquitously expressed in cells and are necessary for vital functions (e.g., transfer RNAs, ribosomal RNA, spliceosome RNA) or are regulatory or selectively expressed at specific developmental stages in certain tissues or in disease (e.g., small ncRNAs, long ncRNAs (lncRNAs), microRNAs) [2].

### 1.1. Small Noncoding RNAs

While the protein-coding genome has undergone limited alterations through evolution, the nonprotein coding genome has significantly changed, and includes classes of small RNAs (sRNAs; 20–200 nucleotides) and lncRNAs (>200 nucleotides) [1,4]. sRNAs are part of all forms of life (i.e., microRNAs and Piwi-interacting RNAs in metazoans; small-interfering RNAs in eukaryotes; small nucleolar RNAs in archaea, bacteria, and eukaryotes).

The most studied type of small ncRNAs are microRNAs (miRNAs) [5]. miRNAs are a family of short non-coding single-stranded RNAs that post-transcriptionally regulate gene expression [6]. miRNAs are 21–25 nucleotides in length. A single miRNA can regulate different functional targets, while multiple miRNAs can target the same mRNA through synergistic or competitive interaction [7]. In animals, the majority of miRNAs target the three prime untranslated region (3’ UTR) and function as translational repressors [8]. miRNAs have roles in numerous biological processes such as cell proliferation, apoptosis, development, differentiation, stress tolerance, energy metabolism and oncogenesis. miRNAs regulate about 30% of human genes, half of which are either tumor-associated or located in sensitive loci [9,10]. In cancers, miRNAs are frequently deregulated through genetic, epigenetic and transcriptional mechanisms, and therefore are correlated with all aspects of the malignancy including regulation of cell cycle, apoptosis, signaling pathways, migration, invasion, immunology and stem cell biology [11]. miRNAs are also involved in cellular response to drugs i.e. chemotherapy [9] and are explored as theranostic tools for brain cancer [5]. Still, the precise molecular mechanism of target gene regulation by miRNAs remains unknown [8].

### 1.2. Long Noncoding RNAs

Conversely, lncRNAs have evolved more recently and are only found in invertebrates and vertebrates, and approximately a third of those known are specific to primates [12]. lncRNAs are found in the developing and adult brain, and they are involved in the promotion of brain development and organization, as well as in transcriptional regulation of neural genes. It is believed that the growth in the number of ncRNAs allowed for the evolution and cellular complexity of the human brain that has led to our higher level of cognition, our behavioral repertoire, and our self-awareness [13].

Long noncoding RNAs (lncRNAs) are highly heterogeneous transcripts, where their lengths can range from at least 200 nucleotides to over 100 kilobases [14]. Although they are not translated to proteins, some have been shown to be involved in the synthesis of small polypeptides [15]. Their lengths also allow them to form intricate structures [16]. According to their localization in the genome, lncRNAs are classified into intronic (i.e., transcribed from introns of protein-coding genes), intergenic (i.e., transcribed from intergenic regions), sense (i.e., transcribed from the sense-strand of protein-coding genes), antisense (i.e., transcribed from the opposite mRNA strand), or bidirectional (i.e., transcribed from the vicinity of sense and anti-sense transcription sites) [14,17]. They can also derive from the mitochondrial genome [13]. lncRNAs are generally localized in the nucleus, have little polyadenylation, and are cell-type and tissue specific [1,18]. The cellular localization of lncRNAs appears to provide information about the mechanisms of cellular regulation [17]. Despite the growing number of lncRNAs defined, information about their functional properties is still limited, which are believed to be related to their secondary structures [19]. As a result of their interactions with DNA, RNA, and proteins [16], lncRNAs are considered to be regulators of gene expression in both health (i.e., cell differentiation, immune responses) and disease (i.e., tumorigenesis). lncRNAs regulate gene expression at the transcriptional, post-transcriptional, and epigenetic levels [20,21]. In cancers, lncRNAs are classified based on the phenotypic characteristics postulated by Hanahan and Weinberg [22,23].

## 2. Glioblastoma

Gliomas are the most prevalent primary brain malignancies. According to the cells of origin they are classified as astrocytomas, oligodendrogliomas, ependymomas, and mixed tumors [3]. The degree of malignancy was initially established in 2007 by the World Health Organization (WHO) classification of tumors of the central nervous system [24], and was complemented with molecular markers in 2016 [25]. This includes histological and genetic information from biopsy specimens, and progression from grade I (benign) to grade IV (malignant glioblastoma).

Glioblastoma is the most lethal of the brain tumors. With standard trimodal therapy, which consists of maximal surgical resection, radiation, and chemotherapy, the average patient survival from diagnosis is 14 months [26], while without treatment, life expectancy is less than 6 months [27]. Tumor progression after surgical resection is also very common, with a fatal outcome expected within 1 year [28]. Disease relapse is believed to be the result of the large intra-tumor and inter-tumor heterogeneity [29,30], which complicates treatment, and also because of the “hypermutator” phenotype that evolves as a result of temozolomide (TMZ) chemotherapy [31,32,33].

In spite of the extensive efforts to understand the nature of glioblastoma, its etiology is still largely unknown. Glioblastoma is characterized by heterogeneity of its molecular and cellular profile and its histological features. Recently, with the development of high-throughput techniques, molecular expression profiling led to the definition of glioblastoma subtypes [34]. Cancer is a complex disease where changes in gene expression contribute to disease appearance, progression, and metastasis. However, as disease complexity goes beyond transcriptomics, new synergistic approaches that include different molecular and epigenetic mechanisms are crucial to our further understanding of the pathogenesis of this disease.

## 3. Long Noncoding RNAs and Their Relationship to Glioblastoma

Despite the advances in clinical management of glioblastoma and implementation of alternative treatment options such as immunotherapies [35,36,37], tumor-treating fields [38,39,40], and gamma knife radiosurgery [38], patient life expectancy has not significantly improved over the last decade. For this purpose, an examination of the less explored molecules like lncRNAs might offer new hope for glioblastoma patients.

### 3.1. Long Noncoding RNAs in Glioblastoma Subclassification and Patient Prognosis

Our understanding of glioblastoma was revolutionized with The Cancer Genome Atlas project, where large amounts of data were analyzed, which led to glioblastoma subclassification [34] and better understanding of the disease. Recently, to obtain a more precise and accurate classification, molecular expression profiles of glioblastoma have been combined with the epigenetic determination of the disease [39,40]. In addition, aberrantly expressed molecular markers have been explored for the identification of glioblastoma subtypes. As an example, differentially expressed lncRNAs have been identified using microarrays based on tissues from glioblastoma patients compared to age-matched donors [19]. lncRNAs regulate gene expression by binding to various histone-modifying enzymes [41]. Moreover, due to their association with chromatin-remodeling complexes, splicing, translation, and protein stability, lncRNAs are also involved in epigenetic regulation [3].

It has been reported that lncRNAs regulate glioblastoma development and malignancy through different cellular processes, including cell proliferation, differentiation, apoptosis, stemness, drug resistance, and response to hypoxia [6,42]. Approximately 40% of known lncRNAs are mainly expressed in the brain [18]. This suggests that lncRNAs have important functions in brain development, and also that changes in noncoding regions and expression can lead to disease-causing variants [43].

Zhang et al. used the Gene Expression Omnibus database to identify sets of lncRNAs specific to different grades of glioma [20]. They analyzed GSE16011 and GSE4290 datasets that contain 425 gliomas and nontumor samples. In the gliomas, colorectal neoplasia differentially expressed nonprotein coding (CRNDE) and LOC400043 were significantly upregulated, while maternally expressed gene 3 (MEG3) was significantly downregulated. Moreover, they showed increased expression of HOX antisense intergenic RNA myeloid 1 (HOTAIRM1) and decreased expression of RFPL1 antisense RNA 1 (RFPL1S) as the malignancy progressed; i.e., in glioblastoma compared to low-grade glioma and the reference samples. By mining The Cancer Genome Atlas and the Gene Expression Omnibus databases, Zhang et al. defined a prognostic glioblastoma signature of six lncRNAs [44]. They showed that the selected six lncRNAs can be used to successfully predict patient survival. Here, patients with high expression levels of KIAA0495 showed shorter survival, while patients with low expression levels of PART1, MGC21881, myocardial infarction-associated transcript (MIAT), growth arrest-specific 5 (GAS5), and Prader Willi/Angelman region RNA 5 (PAR5) showed prolonged survival.

Li et al. performed profiling of glioblastoma lncRNAs using Affymetrix HG-U133 Plus 2.0 arrays, which helped in their subclassification into three novel glioblastoma molecular subtypes with different prognostic values: LncR1, LncR2, and LncR3, enriched in oligodendrocytic signatures (LncR1, LncR3) and astrocytic and neuronal signatures (LncR2, LncR3) [45].

In transcriptomic studies, a reference is needed for the transcripts of interest to be normalized to internal control. Under ideal conditions, the reference should be a molecule of the same RNA class as the target RNA that shows stable expression across different samples under the same experimental conditions. To establish such a “reference lncRNA group”, Kraus et al. examined 90 different lncRNAs in glioma grade II-IV and nontumor tissues [46]. They defined clusters of 24, 22, and 12 lncRNAs that were suitable for normalization for grade II, grade III, and glioblastoma samples, respectively. The glioblastoma lncRNA normalization group consisted of LUCA-15-specific transcript (LUST), small nucleolar RNA host gene 4 (SNHG4), EgoA, H19 upstream conserved 1 and 3 (H19 ups.cons.1,2), EGO B, Zfhx2as, nonprotein coding RNA, upstream of F2R/PAR1 (ncR-uPAR), 21A, HOXA6as, the GAS5-family, neuroblastoma differentiation marker 29 (NDM29), and brain cytoplasmic 200 (BC200).

### 3.2. Long Noncoding RNAs and Their Involvement in Glioblastoma Pathology

In early 2019, Gao et al. published an updated version [47] of the lnc2Cancer database of experimentally supported associations between lncRNAs and different human cancers [48]. The database contains information about experimentally supported (i.e., through RNA interference, in vitro knockdown, western blotting, quantitative PCR, luciferase reporter assays), circulating, and drug-resistant lncRNAs in human cancers from more than 6500 scientific reports. In addition, information about microRNAs and variant regulation, transcription factor binding, and methylation were included. Searching through this database (http://www.bio-bigdata.net/lnc2cancer/) using the keywords “glioblastoma” and “glioblastoma multiforme” (access dates: 31 December 2019; 6 January 2020) provided 91 entries, which are listed in Appendix A. We then searched for lncRNAs that were reported to be drug resistant, circulating, and prognostic, and in addition, which were regulated by methylation, as presented schematically in Figure 1 and as listed in Appendix A. The database also contains a lncRNA cancer-association score that is based on the number of reports that have verified the association and cancer sample type used, i.e., tissue, cell line, blood. For glioblastoma, 91 lncRNAs were obtained, in which H19, metastasis-associated lung adenocarcinoma transcript 1 (MALAT1), LINC00152, taurine upregulated gene 1 (TUG1), and HOX transcript antisense intergenic RNA (HOTAIR) showed the highest cancer-association scores, of 10, 6, 4, 4, and 3, respectively. For an extended list see Appendix A.

Although the current knowledge of the specific functions of lncRNAs remains limited, studies have shown that lncRNAs have the potential to be used as biomarkers for diagnosis and prognosis, and as therapeutic targets for glioblastoma [2,13,26]. In addition, screening of the lncRNA expression profiles of gliomas has shown that there are significant differences between gliomas and reference samples [19,49,50]. Examples of the most commonly reported differentially expressed lncRNAs in glioblastomas are given below.

#### 3.2.1. MALAT

Nuclear MALAT1 (also known as nuclear-enriched abundant transcript 2 (NEAT2)) is among the most abundantly expressed of the lncRNAs in normal tissue. At the molecular level, MALAT1 is recruited to nuclear speckles, and it has been reported to regulate pre-mRNA splicing [51]. MALAT1 was originally described as a prognostic marker for lung cancer metastasis [52]. Its expression is upregulated in lung and breast cancer, colorectal and hepatocellular carcinoma, and downregulated in glioma [52,53]. On the other hand, Xiang et al. showed that expression levels of MALAT1 were a lot higher in glioma tissue than in the para-cancer tissue [54]. A comparison between the U87 and U251 glioma cell lines and normal human astrocytes revealed that the cancer cell lines had higher levels of MALAT1, and it has been recognized as oncogenic in glioma [54,55]. MALAT1 promotes the expression of *sex determining region Y-box 2* (*SOX2*), thus boosting the viability and proliferation of glioblastoma stem cells [56,57], and it also suppresses apoptosis of glioma cells through reduction of Ras-related protein Rap-1b (Rap1B) levels by removal of miR-101 [55]. Additionally, MALAT1 promoted glioma growth in a xenograft mouse model through regulation of miR-129 and *SOX2* [56]. In a prognostic meta-analysis of 40 studies, MALAT1 expression predicted poor overall survival in patients with glioma, with a significant pooled hazard ratio of 2.32 (95% CI, 1.64–3.27; *p* < 0.001) [58]. Ma et al. also showed inverse correlation for the expression levels of MALAT1 and overall survival of patients with glioma, with MALAT1 representing an independent factor for poor prognosis [59].

#### 3.2.2. MEG

The gene for MEG3 is located in human chromosome 14q32.3 within the DLK1-MEG3 locus [60]. Zhang et al. identified a cDNA isoform for MEG3 with no previously known function [61] that strongly suppressed cancer cell growth [62]. However, in their study, MEG3 expression was significantly downregulated in various brain cancers [62] and also in glioma tissue, which negatively correlated with WHO grade in glioma patients [63]. Low MEG3 expression was significantly associated with advanced WHO grade, low Karnofsky performance score, isocitrate dehydrogenase (*IDH*) wild-type, and tumor recurrence, which led to poor overall survival [63]. One of the functions of MEG3 is regulation of the blood–tumor barrier permeability through the MEG3/miR-3305¨Cp/RUNX3 axis [64]. In addition, downregulation of MEG3 was also observed in human glioma cell lines, compared with normal astrocytes. The promoting effects of miR-96-5p on cell growth and metastasis can be reversed by upregulation of MEG3 [65]. Moreover, Wang et al. showed that the expression of MEG3 inhibits cell proliferation and results in G0/G1 arrest in human glioma cell lines through association with p53 [21]. MEG3 promotes autophagy and apoptosis in U251 cells [63] through positive regulation of NAD-dependent deacetylase sirtuin 7 (Sirt7), which is involved in inhibition of the PI3K/AKT/mTOR pathway [66], and through expression of miR-377 [67] and miR-93 [68].

#### 3.2.3. HOTAIR

HOTAIR is a well-characterized lncRNA and its gene is part of the homeobox superfamily located on chromosome 12q13.13 [69]. HOTAIR has critical roles in numerous cancers [70,71,72], it is overexpressed in glioblastoma, and functions in cell proliferation [73]. Tan et al. showed that HOTAIR levels were higher in serum of patients with glioma, and significantly higher in the serum of patients with glioblastoma, compared to the control tissues. The sensitivity and specificity of serum HOTAIR was 86.1% and 87.5%, respectively, with an AUC of 0.913, and may therefore represent a liquid biopsy biomarker for glioblastoma [74]. HOTAIR was also identified through molecular subtyping in glioma, with higher expression in the mesenchymal and classical subtypes than in the neural and proneural subtypes, and as an independent prognostic factor in patients with glioblastoma [75]. Expression levels of HOTAIR > 0.40 had moderate predictive value for grades II-IV glioma. It promotes the invasion of gliomas through upregulation of the expression of matrix metalloproteinase-7 (MMP-7), matrix metalloproteinase-9 (MMP-9), and vascular endothelial growth factor (VEGF) [76]. Integrated in silico analysis, chromatin immunoprecipitation, and quantitative PCR data have shown that homeobox A9 (HOXA9) binds directly to the promoter of *HOTAIR* and might serve as a novel direct regulator of HOTAIR [77]. Inhibition of HOTAIR in an in vivo xenograft mouse model slowed tumor growth, and prolonged the survival of the mice [78]. It has also been shown that combination of the 5′-end of HOTAIR with polycomb repressive complex 2 (PRC2) drives target-gene epigenetic silencing [78,79].

#### 3.2.4. H19

The *H19* gene sequence is located on chromosome 11p15.5, and it was the first human-imprinted noncoding gene to be identified, with expression seen only on the maternal allele. The sequence for lncRNA H19 lies within 200 kbp downstream of the *insulin-like growth factor 2* (*IGF-2*) gene. These genes are imprinted in opposite directions, such that the maternal *H19* and the paternal *IGF-2* alleles are selectively expressed [80]. H19 has a pivotal role in embryogenesis, and fetal growth and development. It accumulates in the human placenta and in several fetal tissues [81], and its expression levels are relatively downregulated after birth [82]. One of the roles of H19 is to contribute to glioblastoma malignancy and to maintain the glioblastoma stem-cell properties. In H19-deficient cells, expression of cancer stem-cell markers Prominin-1 (CD133), homeobox protein NANOG (NANOG), Octamer-binding transcription factor 4 (Oct4), and Sox2 are downregulated [83]. However, expression of H19 is significantly increased in CD133+ cells [84], glioblastoma tissue, and high tumor grade, and is also negatively associated with patient survival [85]. The increased expression of H19 promotes glioblastoma cell migration, angiogenesis, stemness, and tumorigenicity [84]. In cell cultures, H19 promotes glioma-cell growth, invasion and proliferation, and angiogenesis. These tumor-promoting effects of H19 appear to be mediated through miR-675 [85], miR-29a [86], miR-140 [87], and miR-138/HIF-1α [88]. H19-derived miR-675 might regulate glioma-cell proliferation and migration through cyclin dependent kinase 6 (CDK6) and Cadherin 13 (CDH13), and predict poor prognosis for patients with glioma [89,90]. Hypoxia-inducible factor 1-alpha (HIF-1α) directly promotes H19 expression under hypoxia in glioblastoma cells [90]. Downregulation of H19 inhibits cell proliferation, invasion, and migration, and arrests cell-cycle progression. Downregulated H19 also induces cell apoptosis by blocking activation of the Wnt/β-catenin signaling pathway in glioma cells [91,92], and it can lead to epithelial–mesenchymal transition (EMT), with reduced N-cadherin and vimentin [93].

#### 3.2.5. CRNDE

Colorectal Neoplasia Differentially Expressed (CRNDE) was initially associated with colorectal cancer, and its gene is located on chromosome 16 and has oncogenic functions in various cancers. Of all of the lncRNAs, CRNDE shows the highest upregulation in glioma [94]. Szafron et al. showed that increased expression of CRNDE was associated with local invasion, lymph-node metastasis, and poor prognosis for patients with glioma [95]. Four transcript variants of human CRNDE are listed by the National Center for Biotechnology Information, and all of these are upregulated in glioma cells, compared to normal control cells, which suggests that CRNDE has a role in glioma formation [96]. The level of CRNDE expression might modulate the mammalian target of rapamycin (mTOR) signaling pathway [14,96], which is well known for its regulation of cell growth and proliferation. Additionally, CRNDE indirectly inhibits post-transcriptional expression of B-cell lymphoma 2 (Bcl-2) and wingless-type MMTV integration site family, member 2 (Wnt2), which are two downstream effectors of the PI3K/AKT/mTOR signaling pathway that are involved in the malignant characteristics of gliomas, through complete binding to miR-136-5p [97]. Zheng et al. showed high expression of CRNDE in glioblastoma stem cells, which regulated cell proliferation, migration, invasion, and apoptosis. CRNDE binds to and negatively regulates miR-186 [98]. Deletion of CRNDE and overexpression of miR-384 in vivo reduce tumor volume and prolong survival [99]. Moreover, CRNDE expression is a significant independent prognostic factor for overall survival for patients with glioma (*p* = 0.002), is more frequent in advanced glioma, and therefore correlates with larger tumor size and more frequent recurrence [100]. Kiang et al. reported positive association between CRNDE upregulation and epidermal growth factor receptor activation in both glioma tissue and glioma cell lines [101]. Li et al. showed CRNDE involvement in inflammation through the Toll-like receptor pathway, and in the release of transcription factors and cytokines, to promote tumorigenesis and glioma development [102].

In summary, lncRNAs have diverse mechanisms of action in glioblastoma pathology and they act in different ways via different molecular pathways, as presented schematically in Figure 2. However, due to the lack of information about the precise functions of specific lncRNAs, more experimental evidence is needed to confirm their roles in all aspects of glioblastoma pathogenesis, in terms of disease occurrence, resistance to therapy, and disease progression.

## 4. Long Noncoding RNAs and Resistance to Glioblastoma Therapy

### 4.1. Temozolomide and Resistance

As indicated above, chemotherapy with TMZ leads to acquired hypermutation in glioblastoma, where most mutations are present for retinoblastoma (RB) and AKT-mTOR signaling, and promote tumor growth and metastasis [32]. TMZ is a chemotherapeutic drug that is commonly used in the treatment of glioblastoma, as it can prolong patient survival by 2.5 months. In blood, TMZ spontaneously converts to its active form of 5-(3-methyl-1-triazeno)imizadole-4-carboxamide. This is then broken down to produce the methyldiazonium cation, which can methylate DNA at guanines and adenines, to cause fatal damage [103]. The main mechanism that contributes to TMZ resistance is the expression of O^6^-methyl-guanine-DNA methyltransferase (MGMT), which removes the methyl group from the O^6^ of guanine. If the MGMT promoter is methylated, expression of MGMT is reduced, which has been linked to improved patient survival under TMZ treatment [104]. Another factor that greatly contributes to TMZ resistance is the presence of stem-like cells that are particularly resistant to TMZ [105]. Nevertheless, there is growing evidence that TMZ resistance arises through numerous factors, including lncRNAs. The results of these studies are presented schematically in Figure 3, and are outlined in Table 1. They will be further described in the following subsections.

#### 4.1.1. Long Noncoding RNAs Most Commonly Linked to Resistance to Glioblastoma Therapy

As previously described, MALAT1 is one of the most expressed of the lncRNAs in glioblastoma, compared to reference glial cells. Chen et al. analyzed the link between MALAT1 and resistance to TMZ. MALAT1 is overexpressed in both glioblastoma tissue and serum of patients that do not respond to TMZ, in comparison to those that do. TMZ-resistant U251 and U87 cells were developed and MALAT1 was significantly overexpressed in both of these resistant cell lines, in comparison to nonresistant cells. MALAT1 was shown to suppress miR-203, which in turn regulates thymidylate synthase expression [106]. Li et al. similarly analyzed TMZ-resistance mechanisms of MALAT1 in U251 and U87 cells. MALAT1 was overexpressed in TMZ-resistant cells. To determine the mechanism behind this resistance, MALAT1 expression was silenced. In turn, there was reduced expression of multidrug resistance protein 1 (MDR1), multidrug resistance-associated protein 5 (MRP5), low-density lipoprotein receptor-related protein 1 (LRP1), and zinc finger E-box-binding homeobox 1 (ZEB1), a marker of EMT. In addition, expression of epithelial markers E-cadherin and zonula occludens-1 (ZO-1) was increased, and expression of the mesenchymal markers α-smooth muscle actin and fibronectin was decreased. In contrast, overexpression of MALAT1 resulted in increased expression of ZEB1 [107]. Cai et al. also studied the connection between MALAT1 and TMZ resistance. They showed MALAT1 upregulation in glioblastoma in vitro and in vivo. MALAT1 was shown to suppress miR-101 by direct binding. When MALAT1 was decreased in cells, miR-101 was overexpressed and levels of MGMT and Glycogen synthase kinase-3 beta (GSK3B) decreased, concluding that both of these proteins are included in TMZ resistance mechanisms [108].

Another lncRNA highly expressed in glioblastoma, H19, and its association with TMZ resistance was studied by Jia et al. They developed TMZ-resistant U251 and M059J glioblastoma cells and showed that the resistant cells have higher H19 expression than nonresistant cells. When H19 expression was silenced in both of these cell types, the IC_50_ improved from 1000 μM to 600 μM in U251 cells, and from 800 μM to 400 μM in M059J cells. Silencing H19 was also linked to reduced expression of EMT markers and reduced expression of Wnt/β-catenin markers [109]. In further studies on H19, Jiang et al. showed its differential expression in glioblastoma tissue between patients that responded to TMZ and those who did not. There are three variants of H19, and all three of these were present in both of these tissue groups. Variant 1 was significantly overexpressed in TMZ-resistant tissue, whereas there was no difference for the other two variants. H19 was also overexpressed in TMZ-resistant U87 and U251 cells, in comparison to nonresistant U87 and U251 cells. The proposed mechanism of H19 overexpression is linked to the overexpression of common drug resistance genes, such as *MDR*, *MRP,* and *ATP-binding cassette super-family G member 2* (*ABCG2*) [110].

#### 4.1.2. Less Common Long Noncoding RNAs Involved in Resistance to Glioblastoma Therapy

Another lncRNA that has gained attention in terms of TMZ resistance is tumor suppressor candidate 7 (TUSC7), as described by Shang et al. TUSC7 is underexpressed not only in the U87 TMZ-resistant cell line, in comparison to normal U87 cells, but also in glioblastoma tissue from patients who are insensitive to TMZ, in comparison to glioblastoma tissues from TMZ-sensitive patients. In silico analysis revealed that TUSC7 binds miR-10a, which has also been confirmed by RNA pull-down assays. Furthermore, when miR-10a was overexpressed, this led to overexpression of MDR1 [111].

Another lncRNA that appears to regulate miR-10a is RP11-838N2.4, as shown by Liu et al. Microarray analysis of lncRNA in TMZ-resistant U251 and U87 cells revealed that RP11-838N2.4, a transcript of the lncRNA gastric adenocarcinoma-associated, positive CD44 regulator, long intergenic noncoding RNA (GAPLINC), was 7.65-fold downregulated in U87 TMZ-resistant cells in comparison to nonresistant U87 cells. The downregulation was also confirmed by quantitative PCR. It functions as a miR-10a sponge. miR-10a was shown to regulate Ephrin type-A receptor 8 (EphA8). Furthermore, RP11-838N2.4 also regulates transforming growth factor-β signaling pathways independent of miR-10a [112].

Zhao et al. studied the expression of several lncRNAs in secondary TMZ-resistant glioblastoma tissue versus primary glioblastoma tissue. Three samples of each group were included in the study. The microarray analysis showed 299 lncRNA with significance changes in expression levels. Of these, NONHSAT163779 was downregulated and showed the greatest change in expression (8.12-fold) in secondary TMZ-resistant glioblastoma tissue compared to the control primary glioblastoma tissue. NONHSAT163779 regulates the expression of asparagine-linked glycosylation 13 homolog (ALG13), an enzyme involved in glycosylation, which was also downregulated. However, the exact mechanism behind this resistance remains unknown to date [113].

Wu et al. showed that lnc-TALC is overexpressed in TMZ-resistant LN229 cells in comparison to nonresistant LN229 cells. When lnc-TALC was silenced in these TMZ-resistant LN229 cells and in HG7 cells, this led to inhibition of phosphorylation of signal transducer and activator of transcription 3 (Stat3), AKT, and mitogen-activated protein kinase, which implied that lnc-TALC has a role in the tyrosine kinase signaling pathway. Moreover, lnc-TALC binds miR-20-3p, which in turn binds tyrosine-protein kinase Met (MET). The C-Met/Stat3/p300 axis regulates MGMT expression through mediation of acetylation of histone H3 in the MGMT promoter. Expression of lnc-TALC has been shown to be regulated by AKT/FOXO3 [114].

Another lncRNA that has gained attention more recently is ADAMTS9 antisense RNA 2 (ADAMTS9-AS2), which is overexpressed in TMZ-resistant T98G and U228 cells [115]. ADAMTS9-AS2 binds to the RNA-binding protein FUS/TLS (FUS) protein and increases its expression. ADAMTS9-AS2 also decreases the ubiquitination of FUS mediated by E3 ubiquitin-protein ligase (MDM2).

LncRNA P73 antisense RNA 1T non-protein coding (TP73*-AS1*) was described by Mazor et al. as also linked to TMZ resistance, and its expression was shown to be higher in more aggressive tumor types. The expression of TP73*-AS1* was analyzed in the G26 and G7 glioblastoma stem cells. When the expression of TP73*-AS1* was silenced, there was increased cell death. Next, they analyzed the differential expression between the silenced and non-silenced cells, where one of the downregulated mRNAs was aldehyde dehydrogenase 1 family member A1 (ALDH1A1), which is a marker of cancer stem cells [116].

LncRNA AC003092.1 was associated with TMZ resistance by Xu et al. It was 43.99-fold downregulated in the U87 TMZ-resistant cell line, in comparison to the nonresistant cells, and its expression was also lower in glioblastoma tissue from patients who underwent relapse. AC003092.1 was shown to regulate the expression of tissue factor pathway inhibitor 2 (TFPI-2) via miR-195 [117].

Zhang et al. showed elevated expression of SET binding factor 2 antisense RNA 1 (lncSBF2-AS1) in TMZ-resistant cell lines, in comparison to nonresistant cells, and in recurrent glioblastoma tissue, in comparison to primary tissue. In vivo, knockdown of lncSBF2-AS1 resulted in delayed tumor growth for mice treated with TMZ, compared to the control mice. Expression of lncSBF2-AS1 is regulated by the EMT factor ZEB1. As they also confirmed, lncSBF2-AS1 binds miR-151a-3p and the target of miR-151a-3p is XRCC4, which improves the repair of double-strand breaks. Moreover, the exosomes isolated from the TMZ-resistant cell lines contained high levels of lncSBF2-AS1 and promoted TMZ-resistance in TMZ-sensitive cells [118].

### 4.2. Immune-Related Long Noncoding RNAs

Immunotherapy alone and in combination with radiation and chemotherapy is currently explored as a method for glioblastoma treatment [119], but it still needs to prove its clinical importance. The importance of lncRNAs in immune regulation of glioblastomas is also being examined. By regulating the differentiation and function of immune cells, lncRNAs are involved in the adaptive and innate immunity [120]. In glioblastoma, lncRNAs are also correlated to epigenetic regulation of immune response and resistance. They target immune checkpoints and promote formation of immunosuppressive microenvironment, therefore contributing to tumor progression and drug resistance [121]. In the tumor microenvironment, lncRNAs regulate gene expression in response to different stimuli, sponge miRNAs to upregulate expression of immune checkpoints and decrease immuno-surveillance and, at last, encapsulated in exosomes act as mediators of cell–cell communication.

For example, in the study by Li and Meng [122], using data from The Cancer Genome Atlas (TCGA) the authors identified 242 downregulated and 5 upregulated immune-related lncRNAs for lower-grade gliomas. The authors identified 7 lncRNAs (LINC01010, AC135782.1, LINC01711, RFPL1S, LINC02668, AC011899.2, and LINC02192) that can be used as independent prognostic factors.

Moreover, using data from TCGA, Zhou et al. performed genome-wide analysis of the lncRNA expression profiles and clinical information of 419 glioblastoma patients [123]. The authors identified a set of 6 lncRNAs that showed potential to be used as independent prognostic factors either protective (AC005013.5, UBE2R2-AS1, ENTPD1-AS1, and RP11-89C21.2), i.e., their high expression was correlated to longer survival, or risky (AC073115.6 and XLOC_004803), i.e., their high expression was correlated to shorter survival. Additionally, using in silico functional analysis, the authors reported that their prognostic lncRNA signature is enriched in protein-coding genes involved in immune-related biological processes and pathways. At last, the authors showed that the six lncRNA signature was independent of patient age, gender, and tumor subtype. This means that in the future it can help in determining the treatment method of high-risk patients in a pool of patients with the same clinical and molecular characteristics.

In another study, using RNA-seq data from TCGA and Chinese Glioma Genome Atlas (CCGA) Xia et al. identified an immune-related lncRNA signature suitable for predicting survival of glioma patients [124]. The authors identified 812 immune-related lncRNAs associated with glioma. Of these, 11 lncRNAs in particular (H19, DLGAP1-AS1, AC025171.1, PAXIP1-AS2, LINC00205, WDR11-AS1, FEZF1-AS1, THAP7-AS1, HOTAIRM1, HOXD-AS2, and ARHGEF26-AS1) presented as accurate predictors of patient survival.

These findings suggest that specific sets of lncRNAs can be classified as “immune-specific” and alterations in their expression can cause modulation of immune response and surveillance. Still, as this particular field of research is at its early stages, more studies are needed before concrete conclusions can be drawn and lncRNAs can be efficiently clinically utilized.

## 5. Long Noncoding RNAs and Glioblastoma Progression

### 5.1. Glioma Initiation

Due to the aggressive nature of gliomas, informative biomarkers for better histopathological classification and prediction of disease progression are urgently needed. Through molecular genetic analyses, lncRNA gene expression profiles have been determined that have enabled differentiation between glioma and nontumor tissues. Han et al. investigated potential molecular markers using a high-throughput method. In their comparison of glioblastoma tissue with normal brain, they showed 654 upregulated and 654 downregulated lncRNAs. Analysis of the target gene-related pathways revealed significant changes in peroxisome proliferator-activated receptor signaling. As lncRNAs ASLNC22381 and ASLNC20819 were upregulated, these might have important roles in the glioma pathway, and as they probably target IGF-1, they might therefore be associated with recurrence and malignant progression of glioma [19]. In a further study, Chen et al. showed that it is indeed possible to distinguish between different grades of primary and recurrent glioma using lncRNAs. They linked some previously glioma-associated lncRNAs, like H19, CRNDE, and HOTAIRM1, with recurrence, and also defined new lncRNAs, like AC016745.3, XLOC_001711, and RP11-128A17.1, which their lncRNA–mRNA co-expression analysis indicated might have critical roles in glioma recurrence [125].

### 5.2. Glioma Progression

Distinguishing between diseased and normal tissue is of great importance in clinical practice, but as gliomas are tumors of different grades, there is also the need to distinguish between different histological classes. Using gene expression microarray approaches, Zhang et al. and Li et al. performed profiling of glioma lncRNAs, and established the first signatures that provided differentiation of distinct glioma grades [20,45]. Similar studies were performed by other groups. Wang et al. screened several datasets from the Gene Expression Omnibus, the Chinese Glioma Genome Atlas, and the Repository of Molecular Brain Neoplasia Data, and defined a four-lncRNA signature of ArfGAP with GTPase domain, ankyrin repeat and PH domain 2 antisense 1 (AGAP2-AS1), tumor protein translationally controlled 1 antisense 1 (TPT-AS1), long intergenic noncoding RNA 1198 (LINC01198), and MIR155 host gene (MIR155HG). Using this signature, they were able to distinguish between patients with anaplastic glioma who were at high risk, and therefore had more grade IV-like glioma characteristics, and other patients at low risk, with grade II-like glioma. The expression of AGAP2-AS1, LINC01198, and MIR155HG increased with tumor grade, while TPT-AS1 was considered as a protective lncRNA, with greater expression in the low-risk glioma group [126]. Another tumor oncogene, CRNDE, which is also among the most upregulated lncRNAs in glioma, has been associated with tumor progression [100]. Zhi et al. studied lncRNAs in astrocytoma grades II–IV. They defined a potential signature of seven lncRNAs that showed significantly different expression profiles between tumor and normal adjacent tissues. Furthermore, an association with advanced clinical stages of astrocytoma was indicated for upregulation of ENST00000545440 and NR_002809 [127]. Zhen et al. showed that the lncRNA NEAT1 is upregulated in glioma tissue, in comparison to nontumor tissue, and its expression increased with rising pathological grades of gliomas. Further functional analysis determined that NEAT1 functions as a competing endogenous RNA. It binds miR-449b-5p with competitive kinetics and works as a molecular sponge. This leads to upregulation of c-Met, which promotes glioma pathogenesis [128]. ZEB1 antisense RNA 1 (ZEB1-AS1) was shown to be highly expressed in glioma tissues, and to be closely related to glioma clinical stage, which suggested its involvement in glioma progression. An additional functional study of ZEB1-AS1 showed that its silencing might be involved in inhibition of the cell cycle and cell proliferation, migration, and invasion, to thus promote apoptosis of glioma cells [129]. The lncRNA hepatocellular carcinoma upregulated long noncoding RNA (HULC) was initially shown to be highly upregulated in hepatocellular carcinoma, and was shown to have an important role in progression of glioma, which indicated that it might also be a biomarker for survival progression in patients with glioma [130]. Similarly, upregulation of homeobox A11 antisense (HOXA11-AS), homeobox A3 antisense (HOXA-AS3), AB073614, zinc finger antisense 1 (ZFAS1), and sprouty RTK signaling antagonist 4 intronic transcript 1 (SPRY4-IT1) was associated with ascending histological grades, poor prognosis, and increased tumor size [131,132,133,134,135]. On the other hand, cancer susceptibility 2 (CASC2) functions as a tumor-suppressive factor, and its downregulation has been negatively correlated with tumor grade, and poor patient prognosis and survival. Furthermore, investigation of the role of CASC2 here showed that when it was overexpressed, glioma cell proliferation, migration, and invasion were suppressed, through suppression of the Wnt/β-catenin signaling pathway [136]. Similarly, tumor suppressor in lung cancer 1 antisense (TSLC1-AS1) and TUSC7 have been identified as tumor suppressors. Their expression was lower in glioma versus nontumor tissue, and this expression was negatively correlated to tumor grade. Upregulation of both of these lncRNAs showed inhibition of the malignant behavior of glioma cells [137,138]. As angiogenesis is one of the crucial factors that impacts upon tumor development and progression, it is of interest to note that the lncRNA TUG1 has been associated with permeability of the blood–tumor barrier [139]. Indeed, data reported for TUG1 are particularly intriguing; namely, TUG1 has been shown to be upregulated and to act as a molecular switch for vascular endothelial growth factor A [139]. TUG1 is also downregulated in glioma tissues, which negatively correlates with WHO tumor grade, tumor size, and patient survival [140], and suppresses miR-26a, which is particularly amplified in high-grade glioma tissue [141].

In recent studies, the profiling of lncRNA expression was determined also at the epigenetic level. A comprehensive integrative analysis of the DNA methylation status and expression profiles of lncRNAs in different grades of glioma was performed by Li et al., which included 626 patient samples from the Genomic Data Commons Data Portal, with HM450K methylation array data on glioma and glioblastoma from FireBrowse. They showed that lncRNA expression is dynamic during progression of glioma, and they identified 60 lncRNAs that showed significantly differential expression during disease progression. While the lncRNAs identified here included ones for which associations to different types of cancer were already well established, such as HOTAIR, H19, and Pvt1 oncogene (PVT1), they also identified some less-studied lncRNAs, including the Caspase recruitment domain family member 8 antisense RNA (CARD8-AS) and MIR4435-2 host gene (MIR4435-2HG). Furthermore, the epigenetic regulation of the protein-coding and lncRNA genes showed general hypomethylation, with this associated to cancer progression. The majority of the upregulated lncRNAs were hypomethylated, such as PVT1, H19, MIR155HG, Small nucleolar RNA host gene 18 (SNHG18), AC147651.3, SBF2-AS1, and MIR4435-2HG. In all cases here, hypomethylation and upregulation were shown for higher grade glioma. The newly determined MIR4435-2HG might affect the EMT and tumor necrosis factor-α pathways, and might thus act as an oncogenic lncRNA [142]. A further analysis of the expression profiles, functional tests, and epigenetic regulation of lncRNAs was performed by Han et al., and in the publicly available gene expression data they screened, they found 456 differentially expressed lncRNAs between glioblastoma and low-grade glioma. Many of these lncRNAs belonged to the lncRNAs previously characterized as pro-oncogenic lncRNAs, while the MIR22 host gene (MIR22HG) was among the most differentially expressed lncRNAs. Functional analysis revealed that silencing MIR22HG lead to inhibition of the Wnt/β-catenin signaling pathway due to the loss of miR-22-3p and miR-22-5p. Additionally, chromatin immunoprecipitation with parallel DNA sequencing analysis for MIR22HG showed enrichment of acetylation of lysine 27 on histone 3 (H3K27ac), which is a mark of transcriptionally active chromatin and was elevated in glioblastoma tissue, compared to normal brain tissue [143].

As illustrated in Figure 4, the translation of the current knowledge of lncRNAs into clinical practice is already on its way, although with the more recently accumulated knowledge, many new questions have been raised. Careful selection of lncRNAs for determination of their roles in vivo and to understand their underlying molecular mechanisms of action are key features that need to be addressed more deeply to improve our knowledge of glioblastoma, and its development and progression [144].

## 6. Conclusions

Over the past decade, technological progress enabled identification of less known molecules, like lncRNAs, that can have effects on gene expression, and can therefore contribute to biological processes involved in carcinogenesis. Here, we focused on the five most aberrantly expressed lncRNAs in glioblastoma, which have critical roles in tumor initiation and malignant progression. Determination of their specific tissue expression triggered interest in detailed analyses of gene expression signatures. This will enable more accurate molecular differentiation between glioma grades, monitoring of glioma progression and recurrence, and association with poor clinical outcome. With the considerable potential for use of lncRNAs in clinical management of glioblastoma, potential associations between lncRNA profiles and therapy resistance have also been investigated. For the most widely accepted drug treatment of TMZ, the common mechanisms of lncRNAs that impact upon increased expression of EMT markers have been determined. Unveiling new mechanisms of TMZ resistance might identify future targets to decrease drug resistance, and hence to improve the efficacy of TMZ treatment. 

To conclude, we have shown here that lncRNAs are involved in glioblastoma pathogenesis, and that they have roles in regulation of all malignancy aspects. A better understanding of these underlying mechanisms should define lncRNAs as new diagnostic and prognostic biomarkers, and indeed, should also potentiate their use in therapeutic strategies for patients with glioblastoma.

## Figures and Tables

**Figure 1 cancers-12-01842-f001:**
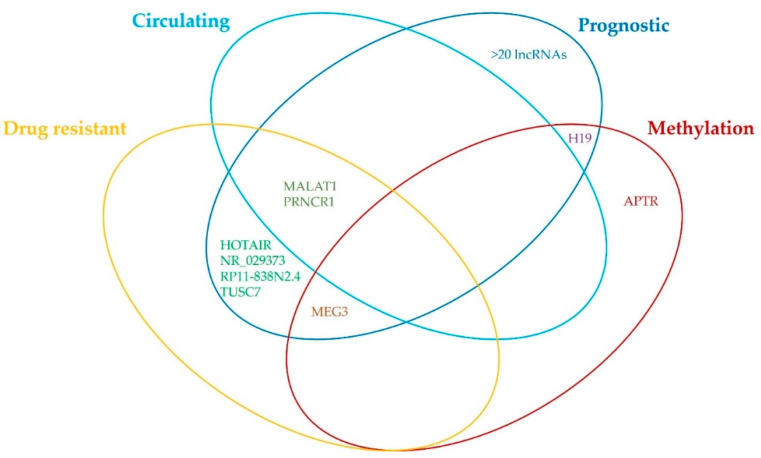
Venn diagram of the findings from lnc2Cancer database about involvement of different lncRNAs in glioblastoma. For more information about the prognostic, drug resistant, circulating, and methylation-regulated lncRNAs please refer to Appendix A.

**Figure 2 cancers-12-01842-f002:**
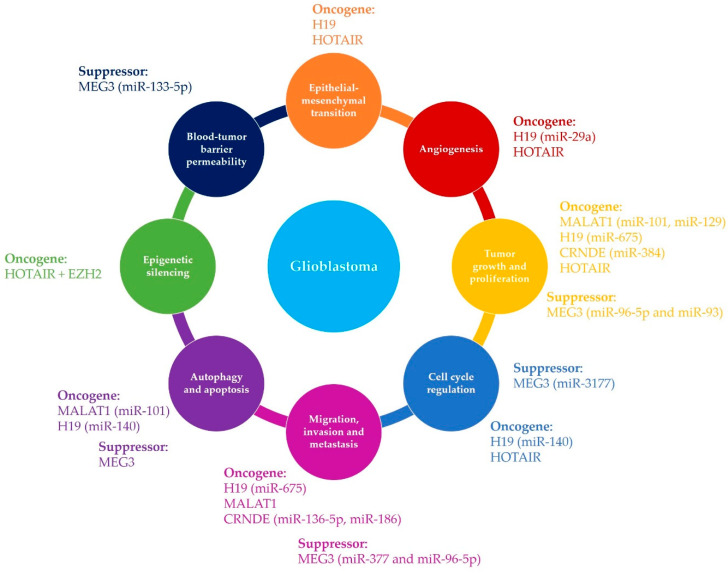
Schematic illustration of the diverse mechanisms by which different lncRNAs are involved in glioblastoma occurrence and progression. lncRNAs contribute to glioblastoma pathogenesis by involvement in different molecular mechanisms including cell proliferation, stemness, angiogenesis, and migration, as well as epigenetic regulation.

**Figure 3 cancers-12-01842-f003:**
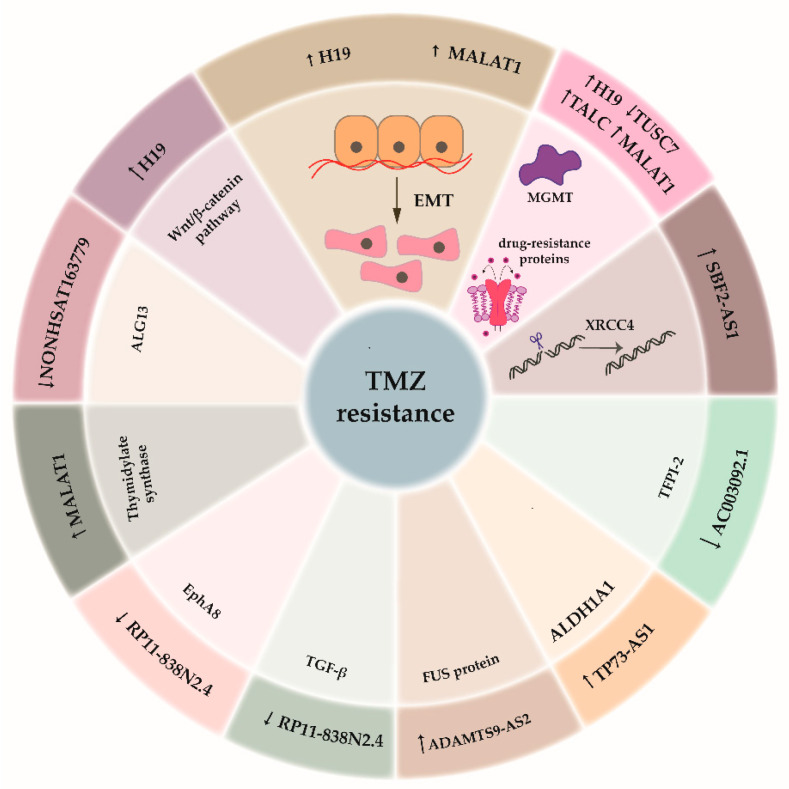
lncRNAs involved in glioblastoma therapy resistance. In the outer shell lncRNAs involved in temozolomide (TMZ) resistance are presented. In the inner shell, the pathways regulated by each specific lncRNA are presented.

**Figure 4 cancers-12-01842-f004:**
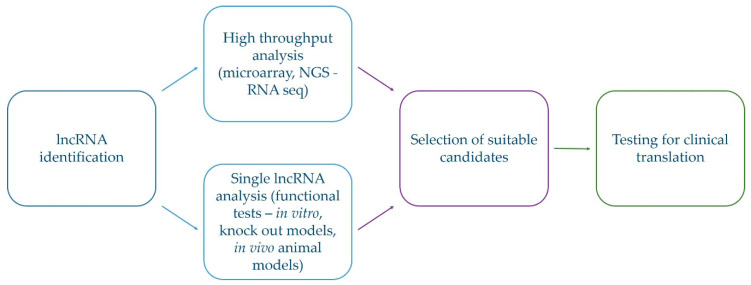
Schematic representation of selection of suitable lncRNA candidates for translation into clinical practice. Selection of promising lncRNAs starts from identification of lncRNAs with differential expression in glioblastoma which is followed by confirmatory tests and examination of the functioning roles. Only after this is completed, their utilization in clinical practice can be considered.

**Table 1 cancers-12-01842-t001:** Most common and less common lncRNAs that are involved in TMZ-resistance.

lncRNA Up- (↑)/Down- (↓) Regulation in TMZ-Resistant Model	Function in Context of Resistance	Model	Reference
↓ NONHSAT163779 ↓ NONHSAT170564 ↓ lnc-NRK-1:2 ↓ lnc-MAP3K9-8:1 ↓ lnc-LOXL3-4:1 ↓ NONHSAT178713 ↓ lnc-COX11-5:1 ↓ ENST00000420774 ↑ NONHSAT154798 ↓ lnc-MYC-16:1 ↓ NONHSAT171282 ↓ T277882 ↑ NONHSAT218984 ↑ NONHSAT211205 ↑ NONHSAT186818 ↓ NR_037403 ↑ T038545 ↑ T350149 ↑ NONHSAT178873 ↑ NONHSAT170013	NONHSAT163779 regulates ALG13, an enzyme involved in glycosylation	Secondary TMZ-resistant glioblastoma	Zhao et al.
↑ H19	Promotes chemoresistance through EMT and Wnt/ß-catenin pathway	TMZ-resistant U251 and M059J	Jia et al.
↑ H19	Linked to overexpression of drug-resistance genes	Glioblastoma tissue from patients TMZ-resistant U87 and U251	Jiang et al.
↑ MALAT1	Regulation thymidylate synthase expression via miR-203	Glioblastoma tissue and serum from patients TMZ-resistant U87 and U251	Chen et al.
↑ MALAT1	Linked to the expression of multidrug-resistance proteins and EMT biomarkers	TMZ-resistant U87 and U251	Li et al.
↑ MALAT1	Regulates miR-101	TMZ-resistant U251	Cai et al.
↓ TUSC7	Regulated MDR1 via binding to miR-10a	Glioblastoma tissue from patients TMZ-resistant U87 glioblastoma	Shang et al.
↓ RP11-838N2.4	Regulates EphA8 via miR-10a Regulates TGF-β signaling pathway independent of miR-10a	Glioblastoma tissue from patients TMZ-resistant U251 and U87	Liu et al.
↑ TALC	Promotes MGMT through regulation of c-Met via mIR-20b-3p	TMZ-resistant LN229, U251, 551W, and HG7	Wu et al.
↑ ADAMTS9-AS2	Binding FUS protein and enhances its stability	Tissue from glioblastoma patients TMZ-resistant T98G and U118	Yan et al.
↑ TP73-AS1	Regulates ALDH1A1	Glioblastoma tissue G26, G7—glioblastoma stem cells	Mazor et al.
↓ AC003092.1	Regulates TFPI-2 via miR-195	Glioblastoma tissue patients TMZ-resistant U87 cell line	Xu et al.
↑ SBF2-AS1	Regulates XRCC4 via miR-151a-3p	U87, LN229, A172, T98, U251, N3, Pri GBM, N3S, Rec GBM, N3T3rd	Zhang et al.

c-Met, Tyrosine-protein kinase Met; XRCC4, X-ray repair cross-complementing protein 4.

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
