# Peer review of "Coding of Glioblastoma Progression and Therapy Resistance through Long Noncoding RNAs"

_cancers, 2020, doi:10.3390/cancers12071842_

Round 1

Reviewer 1 Report

In this manuscript, Alja Zottel et al showed that lncRNAs hold for the development of novel diagnostic and, therpeutic targets that can contribute to prolonged survival and improved quality of life for patients with glioblastoma. The mauscript is written well. The manuscript could be further strengthened with a few additional experiments denoted below. 1. I suggest that the authors need to explain a little bit more about small ncRNAs and microRNAs in part 1 (Long Noncoding RNAs part). 2. It would be more significant if authors make a Schematic illustration (cell signaling pathway) for 4.1 Temozolomide and resistance part. If so, the reader is easily understand the signaling pathway and the information about Temozolomide resistance.

Author Response

In this manuscript, Alja Zottel et al showed that lncRNAs hold for the development of novel diagnostic and, therpeutic targets that can contribute to prolonged survival and improved quality of life for patients with glioblastoma. The mauscript is written well. The manuscript could be further strengthened with a few additional experiments denoted below.

  1. I suggest that the authors need to explain a little bit more about small ncRNAs and microRNAs in part 1 (Long Noncoding RNAs part).

Our reply: We would like to thank the reviewer for this suggestion. Please see newly introduced »1.1. Small Noncoding RNAs« which is now part of the introductory section 1. The Noncoding Genome, pages 1 – 2, lines 36 – 57.

  1. It would be more significant if authors make a Schematic illustration (cell signaling pathway) for 4.1 Temozolomide and resistance part. If so, the reader is easily understand the signaling pathway and the information about Temozolomide resistance.

Our reply: We introduced a new figure to schematically present the currently known relationship between TMZ resistance and lncRNAs. Please see figure 3 on page 8.  

Reviewer 2 Report

i have one minor suggestion

The degree of malignancy was established in 2007 by the WHO. However, in 2016 genetic markers were also included. I think the author should modify their statement. Line 67-71.

Author Response

The degree of malignancy was established in 2007 by the WHO. However, in 2016 genetic markers were also included. I think the author should modify their statement. Line 67-71.

Our reply: Please see revised statement on page 2, lines 86 – 89.

Reviewer 3 Report

Comments-The review article by Zottel et al on “Coding of Glioblastoma Progression and Therapy 2 Resistance through Long Noncoding RNAs” is interesting to read and very well organized. A few minor points that need addressing before final acceptance:

  1. Long sentence. Reframe it and provide missing reference

(Down-regulation of 237 H19 inhibits cell proliferation, invasion, and migration, arrests cell-cycle progression, and induces 238 cell apoptosis by blocking activation of the Wnt/β-catenin signaling pathway in glioma cells [88,89], 239 and it can lead to epithelial–mesenchymal transition (), with reduced N-cadherin and vimentin [90])

  1. There are few linguistic errors throughout the manuscript. Correct them.
  2. Write about antitumor immunotherapeutic relevance of long noncoding RNA in glioblastoma.
  3. Authors have described the role and mechanism of lncRNAs in drug resistance. However, their role in drug resistance with respect to immune invasion is not described. Write few sentences on immune evasion
  4. Authors can take the idea on immune relevance from some of the recent findings-

https://bmccancer.biomedcentral.com/articles/10.1186/s12885-019-6032-3

https://www.ncbi.nlm.nih.gov/pmc/articles/PMC7005925/

https://www.frontiersin.org/articles/10.3389/fonc.2018.00521/full

Author Response

Comments-The review article by Zottel et al on “Coding of Glioblastoma Progression and Therapy Resistance through Long Noncoding RNAs” is interesting to read and very well organized. A few minor points that need addressing before final acceptance:

  1. Long sentence. Reframe it and provide missing reference (Down-regulation of 237 H19 inhibits cell proliferation, invasion, and migration, arrests cell-cycle progression, and induces 238 cell apoptosis by blocking activation of the Wnt/β-catenin signaling pathway in glioma cells [88,89], 239 and it can lead to epithelial–mesenchymal transition (), with reduced N-cadherin and vimentin [90]).

Our reply: Please see revised statement on page 6, lines 268 – 273. We also corrected in the brackets as it was the »EMT« abbreviation that was missing, not a reference.

  1. There are few linguistic errors throughout the manuscript. Correct them.

Our reply: As it is stated in the Acknowledgement section, this manuscript was already edited by a native English-speaking scientific editor. However, we checked it additionally for spelling and grammar errors. If further improvements are needed, please let us know.

  1. Write about antitumor immunotherapeutic relevance of long noncoding RNA in glioblastoma.

Our reply: Please see newly introduced section »4.2. Immune-Related Long Noncoding RNAs« on pages 11 – 12, lines 446 – 484.

  1. Authors have described the role and mechanism of lncRNAs in drug resistance. However, their role in drug resistance with respect to immune invasion is not described. Write few sentences on immune evasion.

Our reply: The literature regarding this matter is very scarce. This information can be found in the section »4.2. Immune-Related Long Noncoding RNAs« on pages 11 – 12, lines 446 – 484.

  1. Authors can take the idea on immune relevance from some of the recent findings

https://bmccancer.biomedcentral.com/articles/10.1186/s12885-019-6032-3

https://www.ncbi.nlm.nih.gov/pmc/articles/PMC7005925/

https://www.frontiersin.org/articles/10.3389/fonc.2018.00521/full

Our reply: We would like to thank the reviewer for providing these manuscripts that served as basis for writing section »4.2. Immune-Related Long Noncoding RNAs«. These findings together with some additional manuscripts are now included in our article.  

Reviewer 4 Report

Great review. I enjoyed reading it from a scientific and clinical point of view. My only question would be to comment on the future of lncRNA in glioblastoma: can they be targeted for treatment? Diagnosis? 

Author Response

Great review. I enjoyed reading it from a scientific and clinical point of view. My only question would be to comment on the future of lncRNA in glioblastoma: can they be targeted for treatment? Diagnosis?

Our reply: We would like to thank the reviewer for the positive evaluation and this question. lncRNA research in glioblastoma is still developing and shows potential to improve the diagnosis of patients. Still, researchers should be very careful as, at this point, most of the data comes from in vitro studies and animal models that must be confirmed in human subjects before clinical translation of the results can be initiated. This can be achieved either with NGS technoglies or experimental validation of human biological samples either tissues or biological fluids. However, it is our professional opinion that lncRNAs should be included in glioblastoma management (both diagnosis and therapy) in combination with the rest of the molecular markers, and should not to be used instead of them or as an independent marker. Because glioblastoma is a complex disease with great heterogeneity, we believe it is important to consider all aspects of the disease in order to get a complete picture of it. When used together with the rest of the molecular and genetic markers, lncRNAs can help in patient stratification and design of alternative therapies for patients who do not benefit from current standard of care.